# Night-to-Day Translation via Illumination Degradation Disentanglement

## Abstract

Night-to-Day translation (Night2Day) aims to achieve day-like vision for nighttime scenes. However, processing night images with complex degradations remains a significant challenge under unpaired conditions. Previous methods that uniformly mitigate these degradations have proven inadequate in simultaneously restoring daytime domain information and preserving underlying semantics. In this paper, we propose **N2D3** (**N**ight-to-**D**ay via **D**egradation **D**isentanglement) to identify different degradation patterns in nighttime images. Specifically, our method comprises a degradation disentanglement module and a degradation-aware contrastive learning module. Firstly, we extract physical priors from a photometric model based on Kubelka-Munk theory. Then, guided by these physical priors, we design a disentanglement module to discriminate among different illumination degradation regions. Finally, we introduce the degradation-aware contrastive learning strategy to preserve semantic consistency across distinct degradation regions. Our method is evaluated on two public datasets, **demonstrating a significant improvement of 5.4 FID on BDD100K and 10.3 FID on Alderley**.

## 1   Introduction

Nighttime images often suffer from severe information loss, posing significant challenges to both human visual recognition and computer vision tasks including detection, segmentation, *etc*. [14]. In contrast, daylight images exhibit rich content and intricate details. Achieving day-like nighttime vision remains a primary objective in nighttime perception, sparking numerous pioneering works [30]. Night-to-Day image translation (Night2Day) offers a comprehensive solution to achieve day-like vision at night. The primary goal is to transform images from nighttime to daytime while maintaining their underlying semantic structure. However, achieving this goal is challenging. It requires to process complex degraded images using unpaired data, which raises additional difficulties compared to other image translation tasks.

Recently, explorations have been made in Night2Day. Early approaches, such as ToDayGAN, demonstrated the effectiveness of cycle-consistent learning in maintaining semantic structure [1]. Subsequent methods incorporated auxiliary structure regularization techniques, including perceptual loss and uncertainty regularization, to better preserve the original structure [33, 18]. Furthermore, some methods utilized daytime images with nearby GPS locations to aid in coarse structure regularization [26]. However, these methods often neglect the complex degradations at nighttime, applying structure regularization uniformly and resulting in severe artifacts. To address this issue, more recent approaches adopt auxiliary human annotations to maintain semantic consistency, such as segmentation maps and bounding boxes [16, 22]. Despite their potential, these methods are labor-intensive and challenging, especially since many nighttime scenes are beyond human cognition.

Submitted to 38th Conference on Neural Information Processing Systems (NeurIPS 2024). Do not distribute.

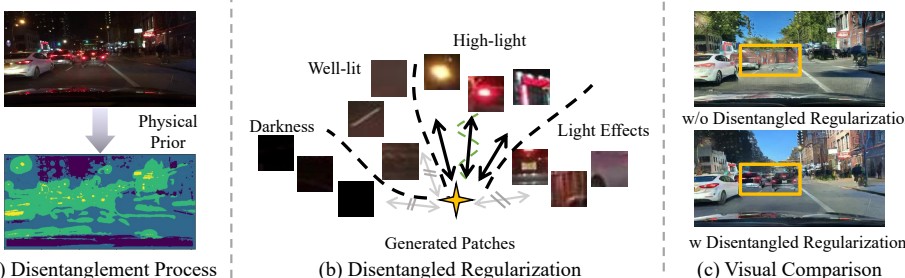

Well-lit
High-light
Darkness
Light Effects
Generated Patches

w/o Disentangled Regularization
w Disentangled Regularization

Physical Prior

(a) Disentanglement Process
(b) Disentangled Regularization
(c) Visual Comparison

Figure 1: Illustration of our motivation. (a) The disentanglement process leverages physical priors. (b) The image patches are restored individually for each degradation type. (c) The proposed Disentangled Regularization improves the overall performance.

The critical limitation of the aforementioned methods is the disregard for complex degraded regions. Specifically, different regions in nighttime images possess varying characteristics, such as extreme darkness, well-lit regions, light effects, *etc*. Treating all these degraded regions equally could adversely impact the results. As illustrated in Figure 1, our key insight emphasizes that nighttime images suffer from various degradations, necessitating customizing restoration for different degradation types. Intuitively, we manage to disentangle nighttime images into patches according to the recognized degradation type and learn individual restoration patterns for them to enhance the overall performance.

Motivated by this point, we propose N2D3 (**N**ight to **D**ay via **D**egradation **D**isentanglement), which utilizes Generative Adversarial Networks (GANs) to bridge the domain gap between nighttime and daytime in a degradation-aware manner, as illustrated in Figure 2. There are two modules in N2D3, including physical-informed degradation disentanglement and degradation-aware contrastive learning, which are employed to preserve the semantic structure of nighttime images. In the disentanglement of nighttime degradation, a photometric model tailored to nighttime scenes is conducted to extract physical priors. Subsequently, the illuminance and physical priors are integrated to disentangle regions into darkness, well-lit, high-light, and light effects. Building on this, degradation-aware contrastive learning is designed to constrain the similarity of the source and generated images in different regions. It comprises disentanglement-guided sampling and reweighting strategies. The sampling strategy mines valuable anchors and hard negative examples, while the reweighting process assigns their weights. They enhance vanilla contrastive learning by prioritizing valuable patches with appropriate attention. Ultimately, our method yields highly faithful results that are visually pleasing and beneficial for downstream vision tasks including keypoint matching and semantic segmentation.

Our contributions are summarized as follows:

(1) We propose the N2D3 translation method based on the illumination degradation disentanglement module, which enables degradation-aware restoration of nighttime images.

(2) We present a novel degradation-aware contrastive learning module to preserve the semantic structure of generated results. The core design incorporates disentanglement-guided sampling and reweighting strategies, which greatly enhance the performance of vanilla contrastive learning.

(3) Experimental results on two public datasets underscore the significance of considering distinct degradation types in nighttime scenes. Our method achieves state-of-the-art performance in visual effects and downstream tasks.

## 2   Related Work

**Unpaired Image-to-Image Translation.** Unpaired image-to-image translation addresses the challenge of lacking paired data, providing an effective self-supervised learning strategy. To overcome the efficiency limitations of traditional cycle-consistency learning, Park *et al.*, first introduces contrastive learning to this domain, achieving efficient one-sided learning[20]. Following this work, several studies have improved the contrastive learning by generating hard negative examples [24], re-weighting positive-negative pairs [31], and selecting key samples [9]. Furthermore, other constraints, such as density [27] and path length [28], have been explored in unpaired image translation. However, all these works neglect physical priors in the nighttime, leading to suboptimal results in Night2Day.

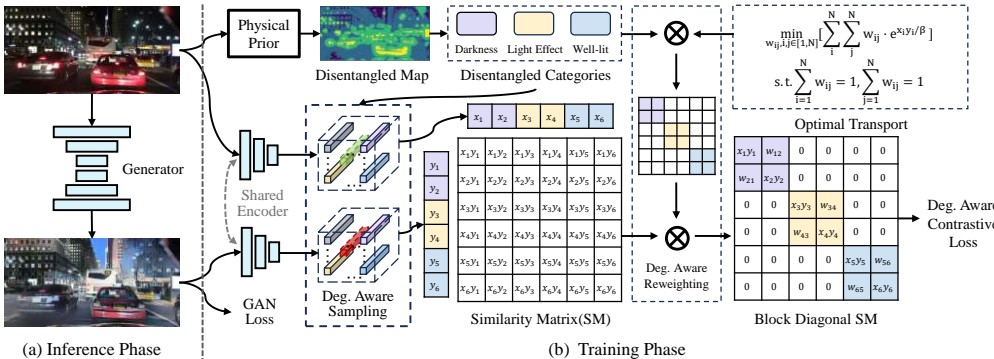

Figure 2: The overall architecture of the proposed N2D3 method. The training phase contains the physical prior informed degradation disentanglement module and degradation-aware contrastive learning module. They are utilized to optimize the ResNet-based generator which is the main part in the inference phase.

**Nighttime Domain Translation.** Domain translation techniques have been applied to address adverse nighttime conditions. An early contribution is made by Anoosheh *et al.*, which demonstrates the effectiveness of cycle-consistent learning in Night2Day[1]. Following this, many works incorporate different modules into cycle-consistent learning to enhance structural modeling capabilities. Zheng *et al.* incorporate a fork-shaped encoder to enhance visual perceptual quality[33]. AUGAN employs uncertainty estimation to mine useful features in nighttime images[18]. Fan *et al.* explore inter-frequency relation knowledge to streamline the Night2Day process[5]. Xia *et al.* utilize nearby GPS locations to form paired night and daytime images, providing weak supervision[26]. Some other studies incorporate human annotations to impose structural constraints, overlooking the practical difficulty of acquiring such annotations at nighttime with multiple degradations [11][16] [22]. To address the concerns of the aforementioned methods, the proposed N2D3 explores patch-wise contrastive learning with physical guidance, so as to achieve degradation-aware Night2Day. N2D3 is free of human annotations and offers comprehensive structural modeling to provide faithful translation results.

## 3 Methods

Given nighttime image $\mathbf{I}_\mathcal{N} \in \mathcal{N}$ and daytime image $\mathbf{I}_\mathcal{D} \in \mathcal{D}$, the goal of Night2Day is to translate images from nighttime to daytime while preserving content semantic consistency. This involves the construction of a mapping function $\mathcal{F}$ with parameters $\theta$, which can be formulated as $\mathcal{F}_\theta : \mathbf{I}_\mathcal{N} \to \mathbf{I}_\mathcal{D}$. Our method N2D3 is illustrated in Figure 2. To train a generator for Night2Day, we employ GANs as the overall learning framework to bridge the domain gap between nighttime and daytime. Our core design, consisting of the degradation disentanglement module and the degradation-aware contrastive learning module, aims to preserve the structure from the source images and suppress artifacts.

In this section, we first introduce physical priors in the nighttime environment, and then describe the degradation disentanglement module and the degradation-aware contrastive learning module, respectively.

### 3.1 Physical Priors for Nighttime Environment

The illumination degradations at night are primarily categorized as darkness, well-lit regions, high-light regions, and light effects. As shown in Figure 3, well-lit represents the diffused reflectance under normal light, while the light effects denote phenomena such as flare, glow, and specular reflections. Intuitively, these regions can be disentangled through the analysis of illumination distribution. Among these degradation types, darkness and high-light are directly correlated with illuminance and can be effectively disentangled through illumination estimation.

As a common practice, we estimate the illuminance map $L$ by utilizing the maximum RGB channel of image $\mathbf{I}_\mathcal{N}$ as $L = \max_{c \in R,G,B} \mathbf{I}_\mathcal{N}^c$. Then k-nearest neighbors [4] is employed to acquire three clusters representing darkness, well-lit, and high-light regions. These clusters are aggregated as masks $M_d$, $M_n$, $M_h$. However, the challenge arises with light effects that are mainly related to

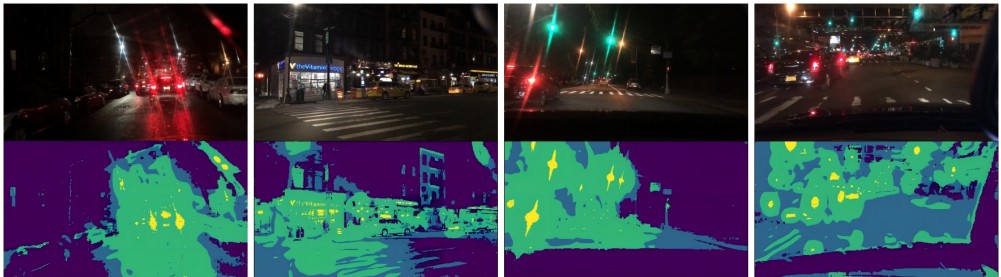

Figure 3: The first row displays nighttime images, while the second row shows the corresponding degradation disentanglement results. The color progression from **blue**, **light blue**, **green** to **yellow** corresponds to the following regions: darkness, well-lit, light effects, and high-light, respectively.

the illumination. Light effects regions tend to intertwine with well-lit regions when using only the illumination map, as they often share similar illumination densities. To disentangle light effects from well-lit regions, we need to introduce additional physical priors.

To extract the physical priors for disentangling light effects, we develop a photometric model derived from Kubelka-Munk theory [17]. This model characterizes the spectrum of light $E$ reflected from an object as follows:

$$E(\lambda, x) = e(\lambda, x)(1 - \rho_f(x))^2 R_\infty(\lambda, x) + e(\lambda, x)\rho_f(x), \tag{1}$$

here $x$ represents the horizontal component for analysis, while the analysis of the vertical component $y$ is the same as the horizontal component. $\lambda$ corresponds to the wavelength of light. $e(\lambda, x)$ signifies the spectrum, representing the illumination density and color. $\rho_f$ stands for the Fresnel reflectance coefficient. $R_\infty$ is the material reflectivity function, formulated as follows at a specific location $x = x_0$:

$$R(\lambda) = a(\lambda) - \sqrt{a(\lambda)^2 - 1}, a(\lambda) = 1 + \frac{k(\lambda)}{s(\lambda)}, \tag{2}$$

where $k(\lambda)$ and $s(\lambda)$ denote the absorption and scattering coefficients, respectively. This formulation implies that for any local pixels, the material reflectivity is determined if the material is given. Assuming $C$ is the material distribution function, which describes the material type varying across locations, the material reflectivity $R_\infty$ can be formulated as:

$$R_\infty(\lambda, x) = R(\lambda)C(x). \tag{3}$$

Since the mixture of light effects and well-lit regions has been obtained previously, the core of disentangling light effects from well-lit regions lies in separating the illumination $e(\lambda, x)$ and reflectance components $R(\lambda)C(x)$. Note that the Fresnel reflectance coefficient $\rho_f(x)$ approaches 0 in reflectance-dominating well-lit regions, while $\rho_f(x)$ approaches 1 in illumination-dominating light effects regions. According to Equation (1), the photometric model for the mixture of light effects and well-lit regions is formulated as:

$$E(\lambda, x) = \begin{cases} e(\lambda, x), & \text{if } x \notin \Omega \\ e(\lambda, x)R(\lambda)C(x), & \text{if } x \in \Omega \end{cases}, \tag{4}$$

where $\Omega$ denotes the reflectance-dominating well-lit regions.

Subsequently, we observe that the following color invariant response to the regions with high color saturation, which is suitable to extract the illumination:

$$N_{\lambda^m x^n} = \frac{\partial^{m+n-1}}{\partial \lambda^{m-1} \partial x^n} \left\{ \frac{1}{E(\lambda, x)} \frac{\partial E(\lambda, x)}{\partial \lambda} \right\}, \tag{5}$$

This invariant has the following characteristics:

$$\begin{aligned} N_{\lambda^m x^n} &= \frac{\partial^{m+n-2}}{\partial \lambda^{m-1} \partial x^{n-1}} \frac{\partial}{\partial x} \left\{ \frac{1}{E(\lambda, x)} \frac{\partial E(\lambda, x)}{\partial \lambda} \right\} \\ &= \frac{\partial^{m+n-2}}{\partial \lambda^{m-1} \partial x^{n-1}} \frac{\partial}{\partial x} \left\{ \frac{1}{e(\lambda, x)} \frac{\partial e(\lambda, x)}{\partial \lambda} + \frac{1}{R(\lambda)C(x)} \frac{\partial R(\lambda)C(x)}{\partial \lambda} \right\} \\ &= \frac{\partial^{m+n-1}}{\partial \lambda^{m-1} \partial x^n} \left\{ \frac{1}{e(\lambda, x)} \frac{\partial e(\lambda, x)}{\partial \lambda} \right\}. \end{aligned} \tag{6}$$

Equation (5) to Equation (6) demonstrate that the invariant $N_{\lambda^m x^n}$ captures the features only related to illumination $e(\lambda, x)$. Consequently, we assert that $N_{\lambda^m x^n}$ functions as a light effects detector because light effects are mainly related to the illumination. It allows us to design the illumination disentanglement module based on this physical prior.

## 3.2 Degradation Disentanglement Module

In this subsection, we will elucidate how to incorporate the invariant for extracting light effects into the disentanglement in computation. As common practice, the following second and third-order components, both horizontally and vertically, are taken into account in the practical calculation of the final invariant, which is denoted as $N$:

$$N = \sqrt{N_{\lambda x}^2 + N_{\lambda \lambda x}^2 + N_{\lambda y}^2 + N_{\lambda \lambda y}^2}. \tag{7}$$

here $N_{\lambda x}$ and $N_{\lambda \lambda x}$ can be computed through $E(\lambda, x)$ by simplifying Equation (5). The calculation of $N_{\lambda y}$ and $N_{\lambda \lambda y}$ are the same. Specifically,

$$N_{\lambda x} = \frac{E_{\lambda x} E - E_\lambda E_x}{E^2}, N_{\lambda \lambda x} = \frac{E_{\lambda \lambda x} E^2 - E_{\lambda \lambda} E_x E - 2 E_{\lambda x} E_\lambda E + 2 E_\lambda^2 E_x}{E^3}, \tag{8}$$

where $E_x$ and $E_\lambda$ denote the partial derivatives of $x$ and $\lambda$.

To compute each component in the invariant $N$, we develop a computation scheme starting with the estimation of $E$ and its partial derivatives $E_\lambda$ and $E_{\lambda \lambda}$ using the Gaussian color model:

$$\begin{bmatrix} E(x,y) \\ E_\lambda(x,y) \\ E_{\lambda \lambda}(x,y) \end{bmatrix} = \begin{bmatrix} 0.06, & 0.63, & 0.27 \\ 0.3, & 0.04, & -0.35 \\ 0.34, & -0.6, & 0.17 \end{bmatrix} \begin{bmatrix} R(x,y) \\ G(x,y) \\ B(x,y) \end{bmatrix}, \tag{9}$$

where $x, y$ are pixel locations of the image. Then, the spatial derivatives $E_x$ and $E_y$ are calculated by convolving $E$ with Gaussian derivative kernel $g$ and standard deviation $\sigma$:

$$E_x(x, y, \sigma) = \sum_{t \in \mathbf{Z}} E(t, y) \frac{\partial g(x - t, \sigma)}{\partial x}, \tag{10}$$

where $t$ denotes the index of the horizontal component $x$ and $\mathbf{Z}$ represents set of integers. The spatial derivatives for $E_{\lambda x}$ and $E_{\lambda \lambda x}$ are obtained by applying Equation (10) to $E_\lambda$ and $E_{\lambda \lambda}$. Then invariant $N$ can be obtained following Equation (8) and Equation (7).

To extract the light effects, ReLU and normalization functions are first applied to filter out minor disturbances. Then, by filtering invariant $N$ with the well-lit mask $M_n$, we obtain the light effects from the well-lit regions. The operations above can be formulated as:

$$M_{le} = \text{ReLU}(\frac{N - \mu(N)}{\sigma(N)}) \odot M_n, \tag{11}$$

while the well-lit mask are refined: $M_n \leftarrow M_n - M_{le}$.

With the initial disentanglement in Section 3.1, we obtain the final disentanglement: $M_d$, $M_n$, $M_h$ and $M_{le}$. All the masks are stacked to obtain the disentanglement map. Through the employment of the aforementioned techniques and processes, we successfully achieve the disentanglement of various degradation regions.

## 3.3 Degradation-Aware Contrastive Learning

For unpaired image translation, contrastive learning has validated its effectiveness for the preservation of content. It targets to maximize the mutual information between patches in the same spatial location from the generated image and the source image as below:

$$\ell(v, v^+, v^-) = -\log \frac{\exp(v \cdot v^+ / \tau)}{\exp(v \cdot v^+ / \tau) + \sum_{n=1}^{Q} \exp(v \cdot v_n^- / \tau)}, \tag{12}$$

$v$ is the anchor that denotes the patch from the generated image. The positive example $v^+$ corresponds to the source image patch with the same location as the anchor $v$. The negative examples $v^-$ represent

patches with locations distinct from that of the anchor $v$. $Q$ denotes the total number of negative examples. In our work, the key insight of degradation-aware contrastive learning lies in two folds: (1) How to sample the anchor, positive, and negative examples. (2) How to manage the focus on different negative examples.

**Degradation-Aware Sampling.** In this paper, N2D3 selects the anchor, positive, and negative patches under the guidance of the disentanglement results. Initially, based on the disentanglement mask obtained in the Section 3.2, we compute the patch count for different degradation types, denoting as $K_s, s \in [1, 4]$. Then, within each degradation region, the anchors $v$ are randomly selected from the patches of generated daytime images $I_{\mathcal{N} \to \mathcal{D}}$. The positive examples $v^+$ are sampled from the same locations with the anchors in the source nighttime images $I_{\mathcal{N}}$, and the negative examples $v^-$ are randomly selected from other locations of $I_{\mathcal{N}}$. For each anchor, there is one corresponding positive example and $K_s$ negative examples. Subsequently, the sample set with the same degradation type will be assigned weights and the contrastive loss will be computed in the following steps.

**Degradation-Aware Reweighting.** Despite the careful selection of anchor, positive, and negative examples, the importance of anchor-negative pairs still differs within the same degradation. A known principle of designing contrastive learning is that the hard anchor-negative pairs (*i.e.*, the pairs with high similarity) should assign higher attention. Thus, weighted contrastive learning can be formulated as:

$$\ell(v, v^+, v^-, w_n) = -\log \frac{\exp(v \cdot v^+/\tau)}{\exp(v \cdot v^+/\tau) + \sum_{n=1}^{Q} w_n \exp(v \cdot v_n^-/\tau)}, \tag{13}$$

$w_n$ denotes the weight of the $n$-th anchor-negative pairs.

The contrastive objective is depicted in the *Similarity Matrix* in Figure 2. The patches in different regions are obviously easy examples. We suppress their weights to 0, which transforms the similarity matrix into a blocked diagonal matrix with $diag(A_1, \ldots, A_4)$. Within each degradation matrix $A_s, s \in [1, 4]$, a soft reweighting strategy is implemented. Specifically, for each anchor-negative pair, we apply optimal transport to yield an optimal transport plan, serving as a reweighting matrix associated with the disentangled results. It can adaptively optimize and avoid manual design. The reweight matrix for each degradation type is formulated as:

$$\min_{w_{ij}, i,j \in [1, K_s]} \left[ \sum_{i=1}^{K_s} \sum_{j=1, i \neq j}^{K_s} w_{ij} \cdot \exp(v_i \cdot v_j^-/\tau) \right],$$
$$\sum_{i=1}^{K_s} w_{ij} = 1, \sum_{j=1}^{K_s} w_{ij} = 1, i, j \in [1, K_s], \tag{14}$$

The aforementioned operations transform the contrastive objective to the *Block Diagonal Similarity Matrix* depicted in Figure 2. As a common practice, our degradation-aware contrastive loss is applied to the $S$ layers of the CNN feature extractor, formulated as:

$$\mathcal{L}_{DegNCE}(\mathcal{F}) = \sum_{l=1}^{S} \ell(v, v^+, v^-, w_n). \tag{15}$$

## 3.4 Other Regularizations

As a common practice, GANs are employed to bridge the domain gap between daytime and nighttime. The adversarial loss is formulated as:

$$\mathcal{L}_{adv}(\mathcal{F}) = ||D(\mathbf{I}_{\mathcal{N} \to \mathcal{D}}) - 1||_2^2,$$
$$\mathcal{L}_{adv}(D) = ||D(\mathbf{I}_{\mathcal{D}}) - 1||_2^2 + ||D(\mathbf{I}_{\mathcal{N} \to \mathcal{D}})||_2^2, \tag{16}$$

where $D$ denotes the discriminator network. The final loss function is formatted as :

$$\mathcal{L}(\mathcal{F}) = \mathcal{L}_{adv}(\mathcal{F}) + \mathcal{L}_{DegNCE}(\mathcal{F}),$$
$$\mathcal{L}(D) = \mathcal{L}_{adv}(D). \tag{17}$$

## 4  Experiments

### 4.1  Experimental Settings

**Datasets.** Experiments are conducted on the two public datasets BDD100K [29] and Alderley [19]. **Alderley** dataset consists of images captured along the same route twice: once on a sunny day and another time during a stormy rainy night. The nighttime images in this dataset are often blurry due to the rainy conditions, which makes Night2Day challenging. **BDD100K** dataset is a large-scale high-resolution autonomous driving dataset. It comprises 100,000 video clips under various conditions. For each video, a keyframe is selected and meticulously annotated with details. We reorganized this dataset based on its annotations, resulting in 27,971 night images for training and 3,929 night images for evaluation.

**Evaluation Metric.** Following common practice, we utilize the *Fréchet Inception Distance* (FID) scores [7] to assess whether the generated images align with the target distribution. This assessment helps determine if a model effectively transforms images from the night domain to the day domain. Additionally, we seek to understand the extent to which the generated daytime images maintain structural consistency compared to the original inputs. To measure this, we employ SIFT scores, mIoU scores and LPIPS distance [32].

**DownStream Vision Task.** Two downstream tasks are conducted. In the Alderley dataset, GPS annotations indicate the locations of two images, one in the nighttime and the other in the daytime, as the same. We calculate the number of SIFT-detected key points between the generated daytime images and their corresponding daytime images to measure if the two images represent the same location. The BDD100K dataset includes 329 night images with semantic annotations. We employ Deeplabv3 pretrained on the Cityscapes dataset as the semantic segmentation model [2], then perform inference on our generated daytime images without any additional training and compute the mIoU (mean Intersection over Union).

Table 1: The quantitative results on Alderley and BDD100k. ↓ means lower result is better. ↑ means higher is better.

| Dataset | | Alderley | | | BDD100k | | |
|---|---|---|---|---|---|---|---|
| Methods | | FID↓ | LPIPS↓ | SIFT↑ | FID↓ | LPIPS↓ | mIoU↑ |
| Original | Conf./Jour. | 210 | - | 3.12 | 101 | - | 15.63 |
| CycleGAN[34] | ICCV 2017 | 167 | 0.706 | 3.36 | 51.7 | 0.477 | 13.42 |
| StarGAN[3] | CVPR 2018 | 117 | - | 3.28 | 68.3 | - | - |
| ToDayGAN[1] | ICRA 2019 | 104 | 0.770 | 4.14 | 43.8 | 0.577 | 16.77 |
| UGATIT[15] | ICLR 2020 | 170 | - | 2.51 | 72.2 | - | - |
| CUT[20] | ECCV 2020 | 64.7 | 0.707 | 6.78 | 55.5 | 0.583 | 9.30 |
| ForkGAN[33] | ECCV 2020 | 61.2 | 0.759 | 12.1 | 37.6 | 0.581 | 11.81 |
| AUGAN[18] | BMVC 2021 | 65.2 | - | - | 38.6 | - | - |
| MoNCE[31] | CVPR 2022 | 72.7 | 0.737 | 6.35 | 40.2 | 0.502 | 17.21 |
| Decent[27] | NIPS 2022 | 76.5 | 0.768 | 6.31 | 40.3 | 0.582 | 10.49 |
| Santa[28] | CVPR 2023 | 67.1 | 0.757 | 6.93 | 36.9 | 0.559 | 11.03 |
| N2D-LPNet[5] | CVPR 2023 | - | - | - | 69.1 | - | - |
| EnlightenGAN [13] | TIP 2021 | 209.8 | - | 2.00 | 103.5 | - | 16.10 |
| Zero-DCE [6] | TPAMI 2022 | 246.4 | - | 4.34 | 90.5 | - | 15.90 |
| DeLight [21] | ECCV 2022 | 222.9 | - | 3.07 | 113.8 | - | 14.48 |
| LLformer [23] | AAAI 2023 | 275.6 | - | 7.62 | 123.1 | - | 15.28 |
| WCDM [12] | ToG 2023 | 239.6 | - | 7.10 | 124.3 | - | 16.32 |
| GSAD [8] | NIPS 2023 | 214.7 | - | 6.29 | 116.0 | - | 15.76 |
| N2D3(Ours) | - | **50.9** | **0.650** | **16.62** | **31.5** | **0.466** | **21.58** |

### 4.2  Results on Alderley

We first apply Night2Day on the Alderley dataset, a challenging collection of nighttime images captured on rainy nights. In Figure 4, we present a visual comparison of the results. CycleGAN [34] and CUT [20] manage to preserve the general structural information of the entire image but often lose many fine details. ToDayGAN [1], ForkGAN [33], Decent [27], and Santa [28] tend to miss important elements such as cars in their results.

In Table 1, thirteen translation methods and three enhancement methods are compared, considering both visual effects and keypoint matching metrics. Our method showcases **an improvement of 10.3**

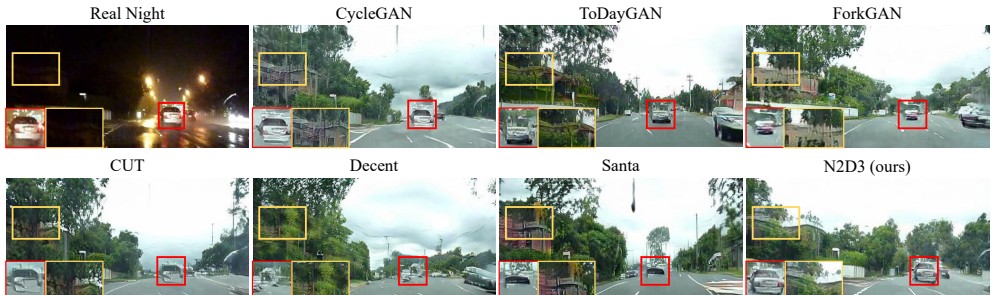

Figure 4: The qualitative comparison results on the Alderley dataset.

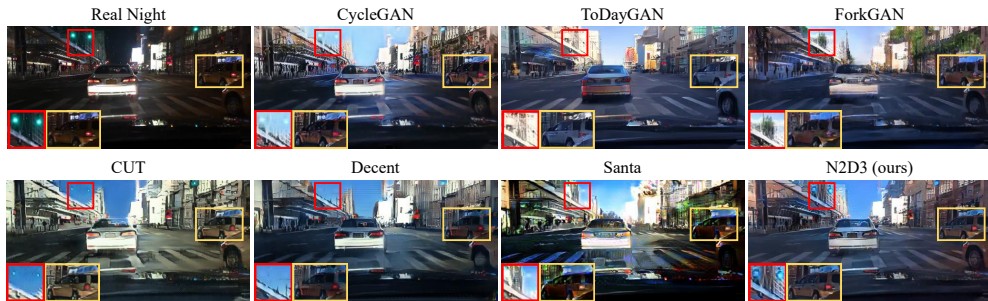

Figure 5: The qualitative comparison results on the BDD100K dataset.

**in FID scores and 4.52 in SIFT scores** compared to the previous state-of-the-art. This suggests that N2D3 successfully achieves photorealistic daytime image generation, underscoring its potential for robotic localization applications. The qualitative comparison results are demonstrated in Figure 4. In conclusion, N2D3 achieves top scores in both FID and LPIPS metrics, demonstrating its superiority in the Night2Day task. N2D3 excels in generating photorealistic daytime images while effectively preserving structures, even in challenging scenarios such as rainy nights in the Alderley.

### 4.3 Results on BDD100K

We conducted experiments on a larger-scale dataset, BDD100K, focusing on more general night scenes. The qualitative results can be found in Figure 5. CycleGAN, ToDayGAN, and CUT succeed in preserving the structure in well-lit regions. ForkGAN, Santa, and Decent demonstrate poor performance in such challenging scenes. Regretfully, none of them excel in handling light effects and exhibit weak performance in maintaining global structures. With a customized design specifically addressing light effects, our method successfully preserves the structure in all regions.

The quantitative results are presented in Table 1. As the scale of the dataset increases, all the compared methods show an improvement in their performance. Notably, N2D3 demonstrates the best performance with **a significant improvement of 5.4 in FID scores**, showcasing its ability to handle a broader range of nighttime scenes and establishing itself as the most advanced method in this domain.

We also investigate the potential of Night2Day in enhancing downstream vision tasks in nighttime environments using the BDD100K dataset. The quantitative results are summarized in Table 1. The enhancement methods demonstrate a slight improvement in segmentation results, while some image-to-image translation methods have a negative impact on performance. N2D3 exhibits the best performance in enhancing nighttime semantic segmentation with **a remarkable improvement of 5.95 in mIoU** compared to inferring the segmentation model directly on nighttime images.

In conclusion, N2D3 achieves top scores in both FID and LPIPS metrics, establishing itself as the most advanced method for the Night2Day task. It excels in generating photorealistic daytime images while preserving local and global structures. Moreover, the substantial improvement in nighttime semantic segmentation highlights its benefits for downstream tasks and its potential for wide-ranging applications.

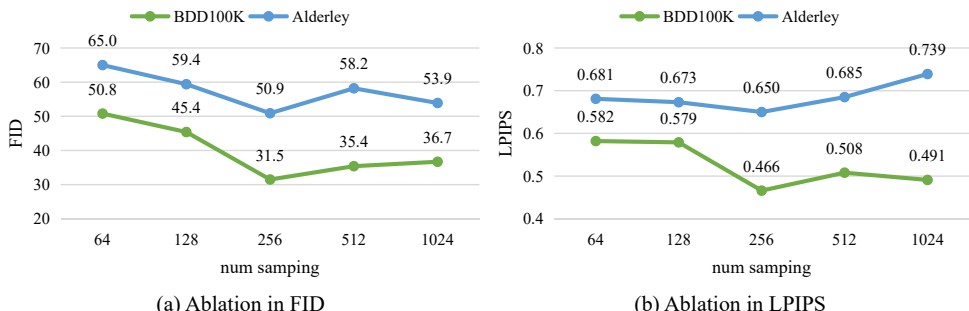

(a) Ablation in FID                                    (b) Ablation in LPIPS

Figure 6: The quantitative results of ablation on the number of patches of the degradation-aware sampling.

Table 2: The quantitative results of ablation on the main component of degradation-aware contrastive learning. (a) denotes the degradation-aware sampling, and (b) denotes the degradation-aware reweighting. $L$ and $N$ denotes the invariant types.

| Main Component | | BDD100K | | Alderley | | | Invariant Type | | BDD100K | | Alderley | | |
|---|---|---|---|---|---|---|---|---|---|---|---|---|---|
| (a) | (b) | FID | LPIPS | FID | LPIPS | SIFT | $L$ | $N$ | FID | LPIPS | FID | LPIPS | SIFT |
| ✗ | ✗ | 55.5 | 0.583 | 64.7 | 0.707 | 6.78 | ✗ | ✗ | 55.5 | 0.583 | 64.7 | 0.707 | 6.78 |
| ✓ | ✗ | 36.9 | 0.495 | 56.6 | 0.698 | 16.52 | ✓ | ✗ | 49.1 | 0.592 | 62.9 | 0.726 | 9.83 |
| ✓ | ✓ | 31.5 | 0.466 | 50.9 | 0.650 | 16.62 | ✓ | ✓ | 31.5 | 0.466 | 50.9 | 0.650 | 16.62 |

## 4.4 Ablation Study

**Ablation on the main component of degradation-aware contrastive learning.** The core design of the degradation-aware contrastive learning module relies on two main components: (a) degradation-aware sampling, and (b) degradation-aware reweighting. As shown in Table 2, when degradation-aware sampling is exclusively activated, there is a noticeable decrease in FID on both datasets compared to the baseline (no components activated). Notably, the combination of degradation-aware sampling and reweighting achieves the lowest FID on both BDD100K and Alderley, indicating the effectiveness of degradation-aware sampling in conjunction with degradation-aware reweighting.

**Ablation on the number of patches in the degradation-aware sampling.** To explore the impact of the number of sampling patches in our method, we conduct an ablation study on the number of sampling patches with settings of 64, 128, 256, 512, and 1024 for degradation-aware sampling. The FID and LPIPS scores are evaluated, as shown in Figure 6. The optimal performance is achieved with 256 patches, and increasing the number of sampling patches beyond this point leads to a degradation in performance.

**Ablation on the type of the invariant in disentanglement.** To explore different invariants for obtaining degradation-disentangled prototypes, we conduct an ablation study on the type of invariant. As shown in Table 2, when $L$ is enabled, the FID decreases from 55.5 to 49.1 on BDD100K and from 64.7 to 62.9 on Alderley. This suggests that incorporating illuminance maps helps in reducing the perceptual gap between generated and source nighttime images. When $N$ is activated, there is a consistent improvement in FID on both datasets, indicating that considering physical priors invariant contributes to more realistic image generation. The combination of both illuminance map and physical prior invariant results in the lowest FID on both datasets, showcasing the complementary nature of these degradation types in improving contrastive learning.

## 5 Conclusion

This paper introduces a novel solution for the Night2Day image translation task, focusing on translating nighttime images to their corresponding daytime counterparts while preserving semantic consistency. To achieve this objective, the proposed method begins by disentangling the degradation presented in nighttime images, which is the key insight of our method. To achieve this, we contribute a degradation disentanglement module and a degradation-aware contrastive learning module. Our method outperforms the existing state-of-the-art, which shows the effectiveness of N2D3 and the superiority of the insight to disentangle the degradation.

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

## A Overview

This supplementary material is organized as follows. Appendix B provides additional details about the proof that the invariant $N_{\lambda^m x^n}$ is exclusively related to the illumination. Appendix C outlines the limitations and failure case of N2D3. Appendix D illustrates the implementation details, including N2D3 and other methods used in the experiments. Appendix E presents additional visualization results.

## B More Proof Details

We provide a detailed proof process to demonstrate **how the invariant $N_{\lambda^m x^n}$ is exclusively related to the illumination and can function as the light effect detector.** First, consider the following equations, corresponding to Equation (5) in the main paper:

$$
\begin{aligned}
N_{\lambda^m x^n} &= \frac{\partial^{m+n-2}}{\partial \lambda^{m-1} \partial x^{n-1}} \frac{\partial}{\partial x} \left\{ \frac{1}{E(\lambda, x)} \frac{\partial E(\lambda, x)}{\partial \lambda} \right\} \\
&= \frac{\partial^{m+n-2}}{\partial \lambda^{m-1} \partial x^{n-1}} \frac{\partial}{\partial x} \left\{ \frac{1}{e(\lambda, x)} \frac{\partial e(\lambda, x)}{\partial \lambda} + \frac{1}{R(\lambda) C(x)} \frac{\partial R(\lambda) C(x)}{\partial \lambda} \right\},
\end{aligned}
\tag{18}
$$

by applying the additivity of linear differential operators, the first term represents the invariants only related to the illumination. The second term can be simplified by applying the chain rule as follows:

$$
\begin{aligned}
&\frac{\partial}{\partial x} \left\{ \frac{1}{R(\lambda) C(x)} \frac{\partial R(\lambda) C(x)}{\partial \lambda} \right\} \\
&= \frac{1}{R(\lambda)^2 C(x)^2} \left( \frac{\partial^2 \{ R(\lambda) C(x) \}}{\partial \lambda \partial x} \cdot R(\lambda) C(x) - \frac{\partial \{ R(\lambda) C(x) \}}{\partial \lambda} \cdot \frac{\partial \{ R(\lambda) C(x) \}}{\partial x} \right) \\
&= \frac{1}{R(\lambda)^2 C(x)^2} \left( \frac{\partial R(\lambda)}{\partial \lambda} \frac{\partial C(x)}{\partial x} \cdot R(\lambda) C(x) - \frac{\partial R(\lambda)}{\partial \lambda} C(x) \cdot R(\lambda) \frac{\partial C(x)}{\partial x} \right) = 0.
\end{aligned}
\tag{19}
$$

Finally, we conclude that the invariant $N_{\lambda^m x^n}$ is **exclusively related to the illumination** and can be formulated as follows:

$$
\begin{aligned}
N_{\lambda^m x^n} &= \frac{\partial^{m+n-2}}{\partial \lambda^{m-1} \partial x^{n-1}} \frac{\partial}{\partial x} \left\{ \frac{1}{E(\lambda, x)} \frac{\partial E(\lambda, x)}{\partial \lambda} \right\} \\
&= \frac{\partial^{m+n-1}}{\partial \lambda^{m-1} \partial x^n} \left\{ \frac{1}{e(\lambda, x)} \frac{\partial e(\lambda, x)}{\partial \lambda} \right\}.
\end{aligned}
\tag{20}
$$

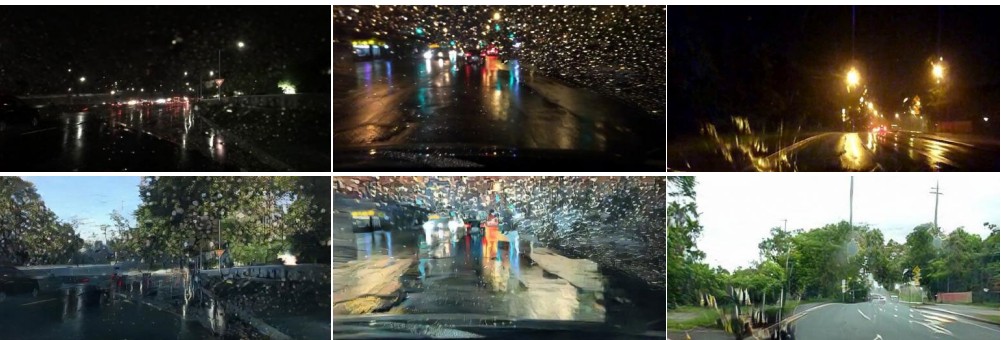

Figure 7: Failure Cases of N2D3: Our method struggles to handle various other types of degradation.

## C Limitations and Failure Case

Despite the superior performance of N2D3 in Night2Day, it still exhibits certain limitations. On the one hand, this work focuses solely on addressing light degradation, while nighttime environments encompass various other types of degradation, including blur caused by rain, motion, and other

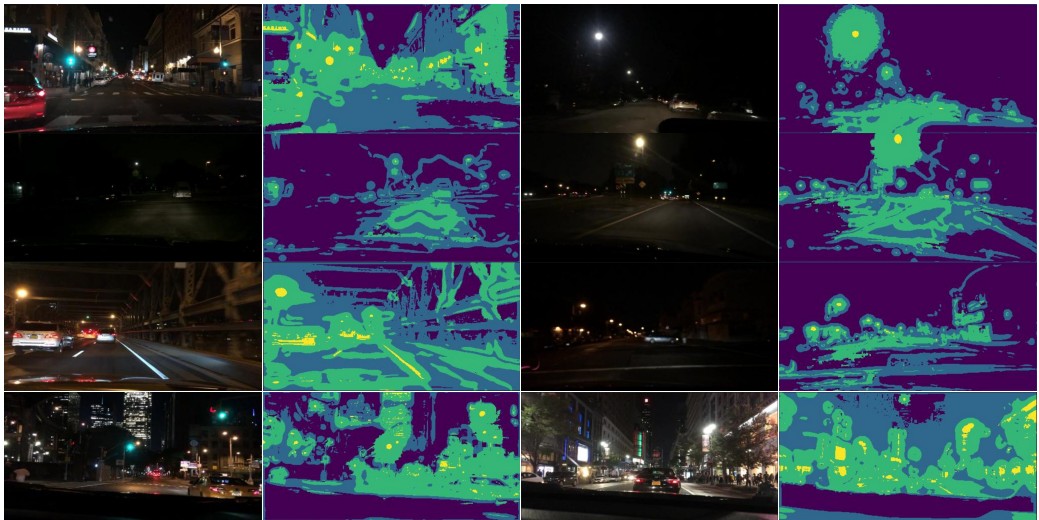

Figure 8: More disentanglement results. The first and third rows display nighttime images, while the second and fourth rows show the corresponding degradation disentanglement results. The color progression from **blue**, **light blue**, **green** to **yellow** corresponds to the following regions: darkness, well-lit, light effects, and high-light.

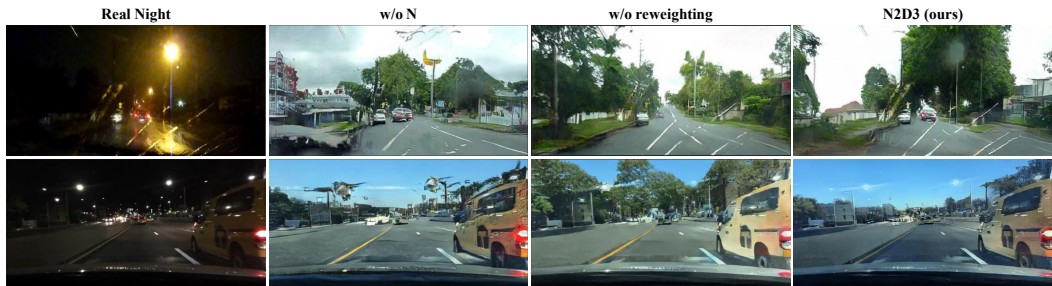

Figure 9: Qualitative comparison abalation results.

factors. Our method currently struggles to handle these situations effectively. On the other hand, the limitations of visible imaging in night vision arise from the scarcity of photos captured in low-light conditions, as illustrated by the failure cases presented inFigure 7. Future advancements in night vision will likely incorporate additional modalities, such as infrared images, radar, and other sensor data, to overcome these challenges and improve performance.

## D  Implementation Details

**Training Details.** We adopt the *resnet_9blocks*, a ResNetbased model with nine residual blocks, as the backbone for generator $G$. Additionally, we utilize the patch-wise discriminator $D$ following PatchGAN[10]. To conduct degradation-aware contrastive learning on multiple layers, we extract features from 5 layers of the generator $G$ encoder, as done in [20]. These layers include RGB pixels, the first and second downsampling convolution, and the first and fifth residual block. For the features of each layer, we apply a 2-layer MLP to acquire final 256-dimensional features. These features are then utilized in our degradation-aware contrastive learning.

All the comparison methods are reproduced using their released source code with default settings. Training procedures are consistent across all methods. All models are trained using the Adaptive Moment Estimation optimizer with an initial learning rate of $10^{-4}$, a momentum of 0.9, and weight decay of $10^{-4}$. For the BDD100K dataset, training consists of 10 epochs with the initial learning rate, followed by another 10 epochs with a decreased learning rate using the polynomial annealing procedure with a power of 0.9. On the Alderley dataset, given the limited training data compared to BDD100K, we extend the training to 20 epochs with the initial learning rate and an additional

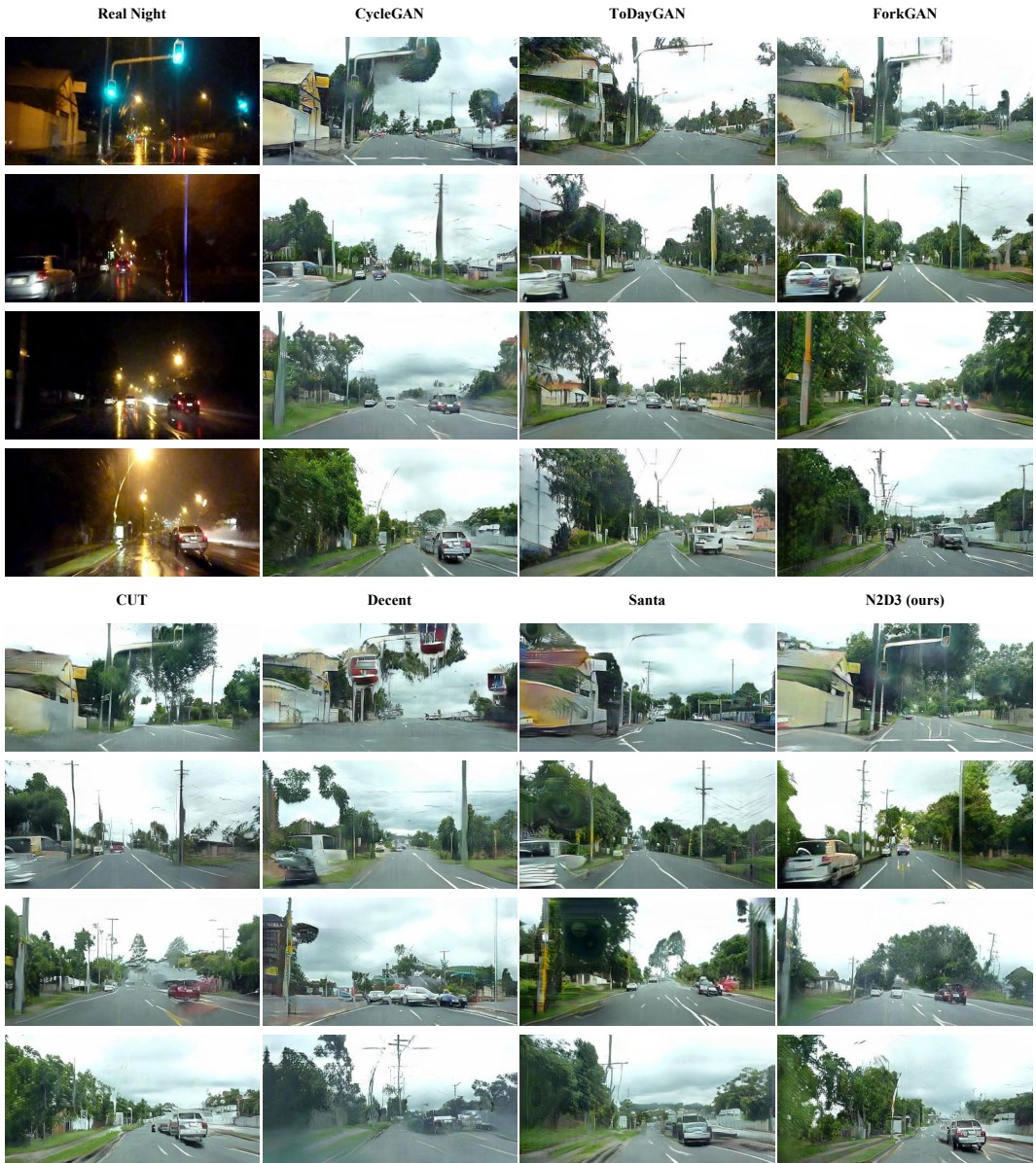

Figure 10: More qualitative comparison results on the Alderley dataset.

20 epochs with the decayed learning rate. All the experiments are run on a single A100 GPU with 80GB of memory. Training our method with a smaller patch size and batch size on a device with less memory is feasible.

**Evaluation Details.** In the evaluation, we compute the *Fréchet Inception Distance* (FID) [7], Structural Similarity Index (SSIM) [25], and Learned Perceptual Image Patch Similarity (LPIPS) [32] scores on $256 \times 512$ images. Partial FID scores are provided by ForkGAN [33], and all SSIM and LPIPS scores are reproduced by us.

Semantic segmentation evaluation are conducted as follows. First, we use Deeplabv3 pretrained on the Cityscapes dataset as the semantic segmentation model [2]. The model is provided by `https://github.com/open-mmlab/mmsegmentation` with an R-18-D8 backbone and trained at a resolution of $512 \times 1024$. Second, we perform $512 \times 1024$ Night2Day translation to obtain the generation results. Finally, we infer the semantic segmentation on the generated daytime images.

# E   More Visualization Results

**More Ablation Visualization Results.** We provide ablation visualization results on both Alderley and BDD100K in Figure 9. The complete method is presented along with ablation studies on the invariant $N$ and without degradation-aware reweighting. All the modules contribute to improving the ability to maintain semantic consistency.

**More Disentanglement Results.** We provide additional disentanglement results in Figure 8. Our disentanglement methods offer a comprehensive representation of different illumination degradation types in various nighttime scenes.

**More Qualitative Comparison.** We present more qualitative comparisons in Figure 10 and Figure 11 alongside other methods. Our method demonstrates visually pleasing results under various nighttime conditions.

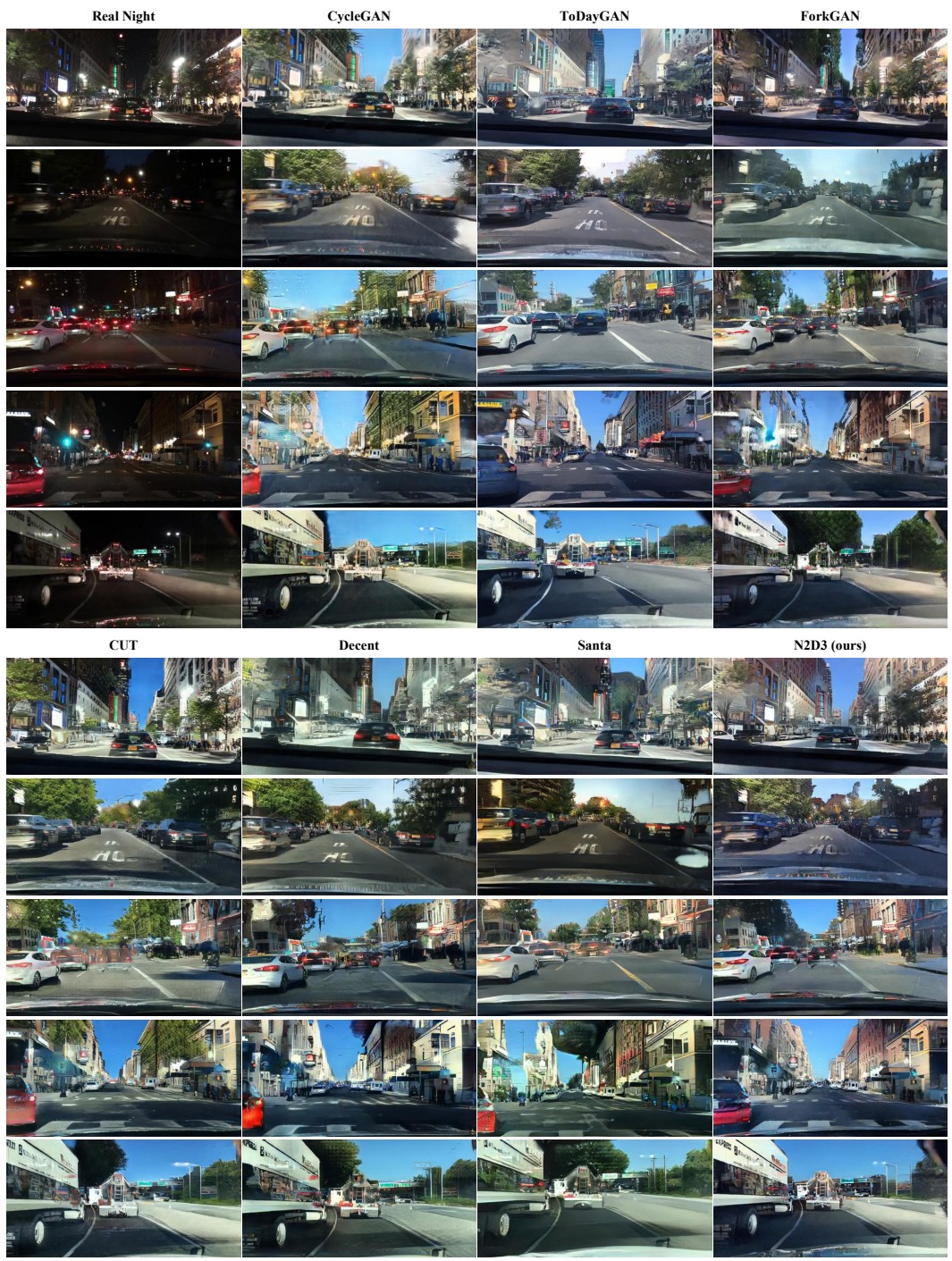

Figure 11: More qualitative comparison results on the BDD100K dataset.

