# OpenReview forum: "Night-to-Day Translation via Illumination Degradation Disentanglement"
_NeurIPS.cc/2024/Conference — Submitted to NeurIPS 2024_

### Official Review · Reviewer_dsJ3 · 2024-06-12

**Soundness:** 2
**Presentation:** 1
**Contribution:** 3
**Rating:** 4
**Confidence:** 4

**Summary:**

This paper presents an approach, namely N2D3, for night-to-day image translation. Specifically, the proposed pipeline involves two stages: illumination degradation disentanglement and degradation-aware contrastive learning. The first stage decomposes an image into darkness, well-lit areas, light effects, and highlight regions. The second stage applies contrastive learning to these four types of nighttime degradations. Extensive experiments conducted on the BDD100K and Alderley datasets demonstrate that N2D3 outperforms existing methods.

**Strengths:**

- The paper addresses the critical problem of night-to-day image translation in computer vision.
- The authors provide comprehensive experimental validation of the proposed method.

**Weaknesses:**

The reviewer has raised this paper for an ethics review due to a significant omission of a key citation. In Section 3.1, the authors introduce a color invariant term for light effect detection. However, this term was originally derived by Geusebroek et al. in their paper *Color Invariance* [1]. The authors devote an entire page to deriving the invariant term without appropriately citing the original work, which violates academic integrity. The authors should explain why this citation is missing, as it does not seem to be an unintentional oversight. This intended missing reference also made Eq. (1)-(5) lack logical coherence and hard to follow.

[1] Color Invariance. J. M. Geusebroek, R. van den Boomgaard, A. W. M. Smeulders, H. Geerts. IEEE TPAMI, 2001.

**Note that although the reviewer has raised the ethics review flag, the reviewer’s rating does not take this into account.**

In addition to the missing citation, the reviewer has concerns about the technical soundness of the paper. Specifically, why are four types of degradation considered? Since the disentanglement of well-lit and light effect regions is the paper’s main contribution, ablation studies using only three types of degradation (darkness, well-lit, and highlight) should be provided.

Besides, the paper’s citation style is inconsistent. For instance, citations for the same conference sometimes include the abbreviation and publisher while others do not (e.g., [1], [19], and [26]). Additionally, some citations include the month of the conference while others do not (e.g., [24], [28]), and some contain volume information while others do not (e.g., [22], [23]). Ensuring consistent citation formatting would enhance the paper’s overall presentation quality.

**Questions:**

Beyond concerns regarding missing citations and technical soundness, the reviewer has the following questions:

- Given the prevalence of large models, why not approach the night-to-day translation task using diffusion models? For instance, a paper [2] addresses day-to-night/fog/snow translation on BDD100K; could this framework be adapted to handle night-to-day tasks? The reviewer does not require quantitative or qualitative results but would appreciate the authors’ insights on this matter.

- What is the rationale behind categorizing degradation into four types? The paper mentions that light effects involve phenomena like flare, glow, and specular reflections. Could degradation be categorized into more types? While a physics-based approach may not apply here, could a segmentation model be trained for this purpose (the reviewer acknowledges the lack of labeled data for training such a model, but there might exist other possible solutions)?

[2] Greenberg et al. S2ST: Image-to-Image Translation in the Seed Space of Latent Diffusion. In CVPR, 2023.

**Limitations:**

The authors have adequately discussed the limitations of the work, and this paper does not have any negative social impacts.

---

> ### Author Rebuttal · Authors · 2024-08-06
>
> ***Q1: Omission of a key citation.***
>
> **R1:**
> We apologize for the omission of this key citation, which has caused confusion. We will ensure that the key citation is included in the revised version.
>
>
> ***Q2: Why are four types of degradation considered?/What is the rationale behind categorizing degradation into four types?***
>
> **R2:**
> At nighttime, the intensity of illumination is the most important criterion for determining whether patterns are far from each other, which categorizes nighttime images into three non-overlapping regions: high-light, well-lit, and darkness. However, within well-lit regions, colored illumination still results in complex patterns with similar intensity levels, which require further subdivision. Therefore, we derive an invariant to extract features related to colored illumination and propose categorizing and disentangling patterns into darkness, well-lit regions, light effects, and high-light to address these challenges.
>
> We hope our explanation can address your concerns.
>
>
> ***Q3: The paper’s citation style is inconsistent.***
>
> **R3:**
> Thank you for your advice. We will revise the citation and ensure the style is consistent in the camera-ready version.
>
> ***Q4: why not approach the night-to-day translation task using diffusion models?***
>
> **R4:**
> Thank you for your advice. We acknowledge that diffusion models have significant potential for addressing translation-based image tasks. However, **applying diffusion models directly does not effectively solve the night-to-day translation problem.** This task is fundamentally a restoration problem that requires recovering information rather than merely performing style transfer, as seen in day-to-night translation tasks. The lack of paired training data further complicates night-to-day translation, making current supervised diffusion backbones insufficient.
>
> While we acknowledge the value and potential of diffusion models for night-to-day translation, **further research into the mechanisms of nighttime imaging and the development of techniques for extracting significant information from such images are more crucial**. These advancements will substantially influence the effective application of diffusion models in night-to-day translation tasks, particularly in selecting controllable information as conditions in the diffusion process.
>
>
> ***Q5: Could degradation be categorized into more types? Could a segmentation model be trained for this purpose?***
>
> **R5:**
> Yes, the degradations can be categorized into more types based on varying levels of illumination density, different types of colored illumination, scattering, and reflective flare. However, apart from degradations with distinct edge structures, such as scattering flares, the efficacy of segmentation models in extracting additional types of degradation is limited.
>
> In my opinion, research into the mechanisms of imaging and optical systems, combined with physically informed machine learning, represents a more promising approach for ultimately addressing night-to-day translation and related nighttime imaging tasks. We sincerely appreciate your advice.

---

> > ### Comment · Reviewer_dsJ3 · 2024-08-08
> > **Response to Author Rebuttal**
> >
> > I have thoroughly read all the reviewers’ comments and the author's rebuttal, and I thank the authors for their responses. However, I still have the following concerns about this paper:
> >
> > **Academic Integrity**: As pointed out by all reviewers, this paper misses a key citation [1], which the authors addressed by “apologizing for our oversight.” This response is insufficient to convince me. The authors acknowledged in their rebuttal to reviewer JHwd that “Eq (5) introduces the invariant from [1],” yet in their response to reviewer rJuz, they wrote “we derive an invariant,” and in their paper, they stated, “we observe that the following color invariant.” Furthermore, in Lines 148-149, the authors wrote, “We develop a computation scheme” and then introduced Eq. (9), which is exactly Eq. (31) in [1]. The authors’ rebuttal only reinforces my suspicion that the missing citation was not an unintentional oversight.
> >
> > [1] Color Invariance. J. M. Geusebroek, R. van den Boomgaard, A. W. M. Smeulders, H. Geerts. IEEE TPAMI, 2001.
> >
> > **Technical Soundness**: The authors argued, "This task is fundamentally a restoration problem that requires recovering information rather than merely performing style transfer.” However, the N2D3 still employs GAN, a generative network, as its backbone, which may introduce semantic changes beyond information recovery. For instance, in Figure 4, the lamppost in the nighttime image disappears in the translated daytime image; in Figure 5, the traffic light appears to be floating in the air.
> >
> > Given these issues, I choose to maintain my initial rating.

---

> ### Author Response · Authors · 2024-08-10
> **Response to Academic Integrity and Technical Soundness**
>
> **Response to Academic Integrity**
>
> Thank you for your valuable comments.
>
> First, we want to assure you that academic integrity is paramount to us. We have taken this feedback seriously and have implemented additional checks in our manuscript preparation process to prevent such oversights in the future. **The omission was due to an unfortunate lapse during our final citation review process.** Citation [1] was included in an earlier version of the manuscript but was inadvertently removed during subsequent revisions, and this omission went unnoticed. Additionally, in Sections 3.1 and 3.2, we mistakenly believed the paper was cited and retained simplified statements, which led to difficulties in understanding. We sincerely apologize for this oversight. We have thoroughly revised our manuscript and now explicitly cite [1] in all relevant sections, including Eq. (5) and Eq. (9).
>
> Second, while our work builds on the invariant presented in [1], **there are several key differences**:
>
> + **The different photometric model.** As the invariant in [1] are derived from the photometric model,$E(λ,x)=e(λ)i(x) R_{\\infty}(λ,x)$, which is tailored for the colored uneven illumination, our invariant are derived from the photometric model :$E(\\lambda, x) = \\{
> \\begin{array}{ll}
> e(\\lambda, x) & \text{if } x \\notin \\Omega,  \\\\
> e(\\lambda, x)R(\\lambda)C(x) & \\text{if } x \\in \\Omega
> \\end{array} $, designed to describe complex illumination in nighttime environments.
>
> + **The different characteristics.**	The different characteristics. The Lemma 8 in [1] introduces the characteristic that extracts object reflectance, specifically edge-relevant features in images. In contrast, the invariant in our work, described in Eq (6), demonstrates the ability of extracting of light effects, which are degradation-related features specific to nighttime conditions
>
> + **The different computation process.** Unlike the computation process in [1], which focuses on extracting high-frequency features such as edges, our work employs two additional normalization and activation functions to extract relatively low-frequency light effects. These steps are specially designed for nighttime disentanglement.
>
> These differences uniquely demonstrate our invariant’s ability to extract light effects in nighttime images, both empirically and theoretically. This specific invariant is derived from our nighttime photometric model, which was developed through our original research.
>
> Despite these distinctions, we fully understand the importance of giving proper credit to prior work. We commit to including citation [1] in the revised version and thank you again for your valuable comments.
>
> [1] Color Invariance. J. M. Geusebroek, R. van den Boomgaard, A. W. M. Smeulders, H. Geerts. IEEE TPAMI, 2001.
>
> **Response to Technical Soundness**
>
> Thank you for your comments.
>
> Enhancement-based methods prioritize preserving information from the night domain. However, **night-to-day translation has higher requirements, prioritizing the conversion of images to the daytime domain first, while maintaining semantic consistency.**
> We acknowledge that the GAN-based methods are not perfect and may result in the semantic changes. To address this, we developed a degradation-aware contrastive learning approach designed to maintain semantic consistency and ensure successful translation. **Our strategy has proven effective, significantly outperforming earlier GAN-based night-to-day works [1, 2, 3].**
>
>
> Additionally, our method demonstrates significant benefits for downstream tasks, such as nighttime image localization and semantic segmentation, compared to enhancement-based methods. As shown in Table 1 of the main paper, **our approach achieves a 5.26 mIoU improvement and a 9-point gain in SIFT scores compared to the most advanced enhancement-based methods**, indicating greater potential for enhancing downstream performance.
>
> Thanks for your comments and we wish these explanations can address your concerns.
>
> [1] Night-to-day image translation for retrieval-based localization. A. Anoosheh, T. Sattler, R. Timofte, M. Pollefeys, and L. V. Gool. ICRA, 2019.
>
> [2] Forkgan: Seeing into the rainy night. Z. Zheng, Y. Wu, X. Han, and J. Shi. ECCV, 2020.
>
> [3] Adverse weather image translation with asymmetric and uncertainty-aware gan. J. Kwak, Y. Jin, Y. Li, D. Yoon, D. Kim, and H. Ko. BMVC, 2021.

---

> ### Comment · Reviewer_dsJ3 · 2024-08-11
> **Further Response to Author Rebuttal**
>
> Thank you to the authors for their further clarification. My concern regarding academic integrity is mostly resolved, though I still find it unusual to use a first-person narrative like "we develop" when referring to results derived from other literature (e.g., Eq. (9)). This phrasing suggests that the person writing the paper may not be the one who proposed the method, conducted the experiments, or fully understood the related work. However, I would now like to focus on new questions that have emerged as I delve deeper into the technical details of this paper.
>
> Firstly, I agree with the authors that their image formation model differs from the one used in [1]. Specifically, [1] assumes a matte, dull surface with Fresnel reflectance set to 1, while this paper employs a model where Fresnel reflectance can either approach 0 or 1. However, I find the derivation from Equation (2) to (3) problematic. Specifically, $R_\infty(\lambda, x)$ cannot be decomposed into two separate functions dependent on $\lambda$ and $x$, respectively.
>
> The explanation following Eq. (2) states that for a given pixel location $x_0$, $R_\infty$ only depends on $\lambda$. While this is obvious since $R_\infty$ depends on two variables $x$ and $\lambda$ only, it does not logically lead to the conclusion that $R_\infty(\lambda, x)$ can be decomposed into $R(\lambda)C(x)$.
>
> For instance, consider a simple model $R(\lambda, x) = \lambda x +1$ (it does not refer to a specific physical model, but it satisfies the statement "for any local pixels, the material reflectivity is determined if the material is given"). If functions $C$ and $R$ exist such that $R_\infty(\lambda, x) = R(\lambda)C(x)$, then setting $x = 0$ would imply $R(\lambda) = \frac{1}{C(0)}$ for all $\lambda$, which is incorrect since $R$ should not be a constant function.
>
> Therefore, further assumptions are necessary for reaching Eq. (3).
>
> Additionally, I have concerns regarding the experiments presented in this paper. Specifically, while the experiments show that using a four-component decomposition is better than a three-component one, they do not demonstrate that this four-component decomposition is optimal or superior to a heuristic approach, such as classifying the pixels into four clusters using a k-nearest algorithm. This ablation is necessary as the qualitative results in Figure (3) seem similar to a simple intensity (RGB) based separation of light-effect and well-lit areas.
>
> Lastly, could the authors clarify what the invariant $L$ represents in Table 2 and Line 278?
>
> Overall, I would like more clarification on Eq. (3) and will reconsider my rating after reviewing the authors’ additional response.

---

> ### Author Response · Authors · 2024-08-12
> **Further Response**
>
> Thank you for your valuable feedback.
>
> **First,** in our photometric model, we assume that the reflectance function $R_{\\infty}(λ,x)$ can be decomposed into the product of two dependent functions: $R(λ)$ and $C(x)$. This decomposition is grounded in the following assumption and definition.
>
> We assume that **materials are uniform and homogeneous within a local area** under normal conditions. Specifically, the optical properties of a material within a small region are described by the function $R(λ)$, which characterizes the material's properties as a function of wavelength and is independent of location. Similar assumption is also used in materials science, optics and computer vision [1, 2, 3]. Under this assumption, we can simplify the reflectivity function $R_{\\infty}(λ,x)$ in the local area to $cR(λ)$, where $c$ is coefficient that describe the material type.
>
> However, this model is limited to describing photometric properties in a local area and does not capture global nighttime conditions. To address this limitation, we introduce the material spatial distribution function $C(x)$ is defined as: $C: \\mathbb{R}^n→{c_1,c_2,…,c_m}$. With $C(x)$, we can model more complex nighttime scenes with diverse material types at macro scales, as detailed in Eq (4) of our paper.
>
> The function $C(x)$ does not influence the derivation of subsequent invariant and their properties. In the derivation of our invariant, the function $C(x)$ appears in both the numerator and denominator of the derivative fraction and cancels out, as shown in Eq (6) of our paper. Nevertheless, the introduction of $C(x)$ is necessary to ensure that our photometric model accurately represents and describes the nighttime environment.
>
> We will emphasize these points in the revised version.
>
>
> [1] Materials science and engineering: an introduction. W. D. Callister and D. G. Rethwisch New York: Wiley, 1999.
>
> [2] Optical properties of solids. E.A. Moore and L.E Smart. CRC Press, 2020.
>
> [3] Color invariance. J. M. Geusebroek, R. van den Boomgaard, A. W. M. Smeulders, H. Geerts. IEEE TPAMI, 2001.
>
> **Second,** we present additional ablation studies on the four components, as detailed in the following tables. The studies reveal that while performance slightly improves with refined classification into four clusters, a more accurate segmentation based on our physical model significantly enhances performance and achieves optimal results. The challenge arises from the similarity in intensity between light effect regions and well-lit areas, making it difficult to differentiate them using a simple KNN. Our physical prior, which extracts features beyond intensity, enables better subdivision and contributes significantly to the final performance.
>
> |                                                                |BDD1000K            |            |       |Alderley      |        |
> | :----------------------------------------  | :--------: | :------:  | :-------: | :---------: | :-------: |
> |                                                                |FID            |LPIPS  |FID           |LPIPS      |SIFT       |
> |3 clusters					       |   49.1     | 0.592 |  62.9     |     0.726     | 9.83 |
> |4 clusters with naïve KNN            	|46.8        |0.529    |60.5    |0.721        | 11.41    |
> |4 clusters with physical prior           |31.5        |0.466    |50.9      |0.650        | 16.62    |
>
> **Third,** in our experiments, $L$ refers to the setting where only three clusters—darkness, well-lit, and high light—based on illumination intensity are used for disentangling. In contrast, $N$ in Table 2 indicates that we further incorporate physical priors to extract light effects during disentangling, using four clusters—darkness, well-lit, light effect, and high light. We will provide additional clarification on this aspect to enhance understanding.

---

> ### Comment · Reviewer_dsJ3 · 2024-08-13
> **Further Response**
>
> Thank you for clarifying the photometric model and notations and providing the additional experiments.
>
> However, as I mentioned under *Technical Soundness,* I believe maintaining semantic consistency is crucial in night-to-day translation, an area where the proposed method falls short. A more straightforward alternative to the GAN-based approach could involve learning a per-pixel curve [1, 2] for different regions identified by N2D3, rather than applying a single curve across the entire image.
>
> [1] Li et al., Learning to Enhance Low-Light Image via Zero-Reference Deep Curve Estimation, TPAMI, 2021.
>
> [2] Wang et al., Unsupervised Face Detection in the Dark, TPAMI, 2023.
>
> Additionally, the intuition behind the 4-category segmentation is not fully explained. While I acknowledge its superiority over 3/4-cluster KNN-based segmentation, I remain concerned about the scalability of ‘Degradation-Aware Contrastive Learning’ when faced with more degradation types, as each patch may contain pixels from multiple categories, and patches of different categories may not necessarily form a *negative* pair. This limitation raises doubts about the method’s foundation for future work.
>
> Lastly, as acknowledged, the paper lacks essential explanations of the photometric model, notations, and important citations.
>
> Given these points, I believe a thorough revision is necessary before acceptance, and I therefore raise my score to a borderline reject.

---

> ### Author Response · Authors · 2024-08-13
> **Response to Reviewer dsJ3**
>
> Thank you for your valuable feedback.
>
> **First,** we recognize that maintaining semantic consistency is crucial in this field. Our methods successfully preserve global semantic consistency across most scenes, though slight artifacts may appear in finer details. Nonetheless, our approach significantly outperforms previous methods in this regard and shows advantages in downstream tasks.
>
> Moreover, our methods are fundamentally different from Zero-DCE [1], as Zero-DCE can not translate a nighttime into the daytime domain (in terms of FID in Table 1), despite GANs are utilized. In downstream tasks, our approach also outperforms Zero-DCE as shown in Table 1.
>
> However, we still acknowledge that introducing per-pixel curves for different degradation types could be a valuable direction for future research. We appreciate your suggestion and will consider it in our ongoing work.
>
> **Second,** for the two misunderstandings that have caused the concerns about the scalability, we would like to make the following clarifications:
>
> Regarding the misunderstanding  about multiple categories, **we clarify that each patch in our method originates from a single category, with no overlap between different degradation types.** Categories such as darkness, well-lit, and high light are uniquely labeled based on illumination intensity using KNN. The light effect regions are decomposed from well-lit regions, as detailed in Eq (11) of our paper, and the refinement operation $M_n←M_n-M_le$ in line 158 ensures no overlap.
>
> Regarding the misunderstanding that patches from different categories may not necessarily form negative pairs, **we clarify that negative pairs are formed within the same category, not across different categories.** This ensures that all negative examples are hard negative examples, meaning they are sufficiently similar to the query patch but not identical. Mainstream theoretical studies indicate that hard negative example mining can enhance the performance of contrastive learning, which is consistent with our empirical results. Our method demonstrates performance advantages compared to randomly selects patches as negative samples (CUT [2]), indicating that our approach effectively mines negative samples.
>
>  Moreover, to further mitigate potential disentanglement errors, we introduced a reweight matrix based on optimal transport, which reassesses the weights for the sampled pairs to ensure optimal negative sample mining. The quantitative comparison in Table 2 confirms that our disentanglement approach significantly enhances performance through effective hard negative sample mining, with additional improvements achieved through the reweighting operations.
>
> **Third,** we have provided a detailed explanation of these aspects in our previous comments and appreciate your recognition of our earlier explanations. In the revised version, we will add the key citations in line 114, and include the assumptions and notations between Eq (2) and Eq (3).
>
> [1] Learning to enhance low-light image via zero-reference deep curve estimation. Li et al., TPAMI, 2021.
>
> [2] Contrastive learning for unpaired image-to-image translation. T. Park. A. A. Efros, R. Zhang, J. Y. Zhu. ECCV 2020.

---

> > ### Comment · Reviewer_dsJ3 · 2024-08-14
> >
> > Thank you for your prompt response.
> >
> > Regarding Zero-DCE, I want to highlight that some recent work [1] has already applied this curve in reverse for the night-to-day translation task. The authors might consider trying this approach.
> >
> > [1] Luo et al., ‘Similarity Min-Max: Zero-Shot Day-Night Domain Adaptation,’ ICCV 2023.
> >
> > Regarding scalability, I apologize for the earlier oversight in selecting negative samples. However, I still have concerns about the patch sampling strategy, as segmentation is based on pixels, which could lead to issues with patches along the borders.
> >
> > I recognize that this paper is borderline after a thorough discussion. However, I am inclined to reject it due to its poor presentation (inconsistent citation style, missing key citations, lack of theoretical assumptions, unclear notations, etc.). While I would recommend a major revision if this were a journal submission, as a conference paper, I have to lean toward a borderline rejection.
> >
> > Thank you again to the authors for their detailed rebuttal. I believe I have fulfilled my responsibilities as a reviewer, even though I have been on vacation and traveling since last week. I will not oppose the AC if they believe the reasons to accept outweigh those for rejection.

---

### Official Review · Reviewer_PUPw · 2024-06-30

**Soundness:** 3
**Presentation:** 3
**Contribution:** 2
**Rating:** 5
**Confidence:** 3

**Summary:**

The paper proposes a new framework N2D3 for solving night to day image translation problem. Their framework consists of a physics-based disentanglement module and a contrastive learning module for preserving semantic consistency. Their method shows improved performance in terms of FID and downstream task performance on BDD100K and Alderley dataset.

**Strengths:**

- Using the Kubelka-Munk theory for different degradation types and applying a patch-based image translation is a novel method.
- The figures are well-made. For instance, the visualization in Fig. 1 and Fig. 2 are intuitive and helpful for understanding the whole architecture.
- Quantitative evaluation results are convincing, showing the effectiveness of the proposed framework in terms of various metrics.

**Weaknesses:**

- Clarity of the method sections can be improved. For instance, including more rigorous definitions or visualizations of what well-lit and different light effects mean and provide a motivation why it is helpful to disentangle those illumination causes separately.
- The authors can also add proper citations to previous work when they mention “by common practice”, for instance in line 107, line 142 and line 196.

**Questions:**

- Why is illuminance computed by the maximum of RGB channel (line 107)? I think this is indeed not a common practice.
- Could you provide some ablation visualizations with and without applying the disentanglement method in 3.2? For instance, the 3 clusters with initial disentanglement in 3.1 and the four clusters afterwards.
- Since the evaluation metrics are performed on segmentation tasks, can you provide some visual examples of the segmented regions of those previous methods and your method?

**Limitations:**

As the authors already mentioned in the appendix, the current physics-aware degradation disentanglement module is designed mostly for illumination related effects and does not handle other types of degradation such as raindrops. I wonder how the authors think the framework could benefit or inspire other types of adverse weather image restoration tasks.

---

> ### Author Rebuttal · Authors · 2024-08-06
>
> ***Q1: Clarity of the method sections can be improved. For instance, including more rigorous definitions or visualizations of what well-lit and different light effects mean and provide a motivation why it is helpful to disentangle those illumination causes separately.***
>
> **R1:** Thanks for the advise.
> A key observation is that, when perform Night2Day translation, treating all regions equally leads to significant artifacts by mixing different patterns from various regions. For example, mixing light effects patterns with high-light patterns, as shown in the Figure 1 of the main paper. Intuitively, separating these patterns and regularizing the structure will help mitigate these artifacts.
>
> Based on this intuition, we sought a disentanglement strategy to separate these patterns. At nighttime, the intensity of illumination is the most important criterion for determining whether patterns are far from each other, which categorizes nighttime images into three non-overlapping regions: high light, well-lit, and darkness. However, within well-lit regions, colored illumination still results in complex patterns with similar intensity levels, which require further subdivision. Therefore, we derive an invariant to extract features related to colored illumination and propose categorizing and disentangling patterns into darkness, well-lit, light effects, and high-light.
>
>  We hope our explanation can address your concerns.
>
> ***Q2: The authors can also add proper citations to previous work when they mention “by common practice”, for instance in line 107, line 142 and line 196.***
>
> **R2:** Thanks for the advice. We will revise these problems in the camera-ready version.
>
> ***Q3: Why is illuminance computed by the maximum of RGB channel (line 107)? I think this is indeed not a common practice.***
>
> **R3:** This estimation operation is broadly used in low-light image enhancement work for rough illumination estimation, such as in LIME (TIP 2016), URetinex-Net (CVPR 2022), and PairLIE (ICCV 2023).  We hope these reference can solve your concerns.
>
> ***Q4: Could you provide some ablation visualizations with and without applying the disentanglement method in 3.2?***
>
> **R4:** We provide such ablation visualization in the PDF file of the Author Rebuttal. Thanks for your advice again.
>
> ***Q5: Can you provide some visual examples of the segmented regions of those previous methods and your method?***
>
> **R5:** We provide visual examples of the segmented regions comparison in the PDF file of the Author Rebuttal. Thanks for your advice again.
>
> ***Q6: I wonder how the authors think the framework could benefit or inspire other types of adverse weather image restoration tasks.***
>
> **R6:** The core of this framework is utilizing physical priors to disentangle the mixture of complex patterns in degraded images, allowing the generator to learn these patterns more effectively. From a metric learning perspective, this approach acts as physical-informed hard negative example mining during contrastive learning. We believe that this disentanglement framework can also benefit other types of adverse weather image restoration with translation-based methods, as it leverages related physical priors to extract corresponding degradation patterns.

---

> > ### Comment · Reviewer_PUPw · 2024-08-11
> > **thank you for the response**
> >
> > I appreciate the authors' response and the additional visualizations provided during the rebuttal. The proposed disentanglement ideas for addressing the night-to-day image translation task is supported by extensive quantitative experiments. While I remain slightly positive about this work, I believe that the theoretical derivations require further development as suggested by other reviewers. Therefore, I will maintain my original rating.

---

> > > ### Author Response · Authors · 2024-08-13
> > > **Response to the Theoretical Derivations**
> > >
> > > Thank you for your valuable feedback and affirmation of our work. We hope that the following clarification in the theoretical derivation addresses your concerns.
> > >
> > > First, we assume that **materials are uniform and homogeneous within a local area** under normal conditions. Specifically, the optical properties of a material within a small region are described by the function $R(λ)$, which characterizes the material's properties as a function of wavelength and is independent of location. Similar assumption is also used in materials science, optics and computer vision [1, 2, 3]. Under this assumption, we can simplify the reflectivity function $R_{\\infty}(λ,x)$ in the local area to $cR(λ)$, where $c$ is coefficient that describe the material type.
> > >
> > > Then we find that this model is  limited to describing photometric properties in a local area and does not adequately capture global nighttime conditions. To address this limitation, we introduce the material spatial distribution function $C(x)$ is defined as: $C: \\mathbb{R}^n→{c_1,c_2,…,c_m}$. With $C(x)$, we can model more complex nighttime scenes with diverse material types at macro scales, as detailed in Eq (4) of our paper.
> > >
> > > The function $C(x)$ does not influence the derivation of subsequent invariant and their properties. In the derivation of our invariant, the function $C(x)$ appears in both the numerator and denominator of the derivative fraction and cancels out, as shown in Eq (6) of our paper. Nevertheless, the introduction of $C(x)$ is necessary to ensure that our photometric model accurately represents and describes the nighttime environment.
> > >
> > > We will emphasize these points in the revised version for better understanding.
> > >
> > > [1] Materials science and engineering: an introduction. W. D. Callister and D. G. Rethwisch New York: Wiley, 1999.
> > >
> > > [2] Optical properties of solids. E.A. Moore and L.E Smart. CRC Press, 2020.
> > >
> > > [3] Color invariance. J. M. Geusebroek, R. van den Boomgaard, A. W. M. Smeulders, H. Geerts. IEEE TPAMI, 2001.

---

### Official Review · Reviewer_rJuz · 2024-07-12

**Soundness:** 2
**Presentation:** 3
**Contribution:** 3
**Rating:** 4
**Confidence:** 5

**Summary:**

This paper presents a comprehensive solution for Night2Day image translation by leveraging physical priors, photometric modeling, and contrastive learning, leading to state-of-the-art performance in visual quality and downstream vision tasks.

**Strengths:**

The authors develop a photometric model based on Kubelka-Munk theory to extract physical priors from nighttime images. This model helps to disentangle different types of illumination degradations by analyzing the illumination distribution.
Overall, the paper presents a novel approach to handling nighttime image translation by considering the unique challenges posed by varying degradations and employing both physical modeling and advanced learning strategies to address these challenges effectively.

**Weaknesses:**

1. The writing is difficult to understand. The explanations and derivations for Eqs (1) to (5) lack logical coherence and necessary references, making them hard to follow.   The derivations for Eqs (7) to (9) also lack supporting references, casting doubt on their validity.
2. The motivation for DAR is unclear. Please explain the motivation behind it.
3. There is no baseline network, making it difficult to determine the performance gain for the specific module.
4. The ablation experiments lack in-depth analysis. For instance, there are no ablation experiments to verify the impact of introducing four regions for disentanglement versus three regions (e.g., excluding the light effects region).
5.There is a need to compare with more recent methods for unpaired image-to-image translation, such as COCO-FUNIT, StegoGAN, GP-UNIT, etc. Please check the reference [5], as it does not seem to be published in CVPR.

**Questions:**

1. The explanations for Eqs (1) to (5) lack logical coherence and necessary references, making them hard to follow. What is the "color invariant response" in Eq (5), and why is it used to extract illuminance? The derivations and calculations for Eqs (7) to (9) also lack supporting references, casting doubt on their validity. Why is there a need to refine Mn?
2. How is the reweighting matrix obtained through optimal transport? Are there any references that have implemented this approach?
3. Contrastive learning has been widely used in image translation. The paper lacks ablation experiments to support the performance gains from the reweighting operation.

**Limitations:**

The motivation behind some methodological choices, is not clearly explained.
The ablation experiments are insufficient and lack depth.
The paper lacks a discussion on the computational complexity and resource consumption of the proposed method.

---

> ### Author Rebuttal · Authors · 2024-08-06
>
> ***Q1: The writing is difficult to understand. The explanations and derivations for Eqs (1) to (5) lack logical coherence and necessary references, making them hard to follow. The derivations for Eqs (7) to (9) also lack supporting references, casting doubt on their validity.***
>
> **R1:** We apologize for the omission of a key citation, which has led to poor readability in this part, as explained in the author rebuttal. Moreover, Eq (1) is derived from [1], and Eqs (2)–(4) are the simplified photometric model tailored for nighttime environments proposed by us. Eq (5) presents the invariant from [1], and we derived new characteristics based on our simplified model in Eq (6). From Eqs (7)–(9), we follow [1] to compute the basic invariant and refine it for disentangling. We hope this explanation address your concerns.
>
> [1] Color Invariance. J. M. Geusebroek, R. van den Boomgaard, A. W. M. Smeulders, H. Geerts. IEEE TPAMI, 2001.
>
> ***Q2:  The motivation is unclear. Please explain the motivation behind it.***
>
> **R2:** A key observation is that, when perform Night2Day translation, treating all regions equally leads to significant artifacts by mixing different patterns from various regions, as shown in the Figure 1 of the main paper. Intuitively, separating these patterns and regularizing the structure will help mitigate these artifacts. At nighttime, the intensity of illumination categorizes nighttime images into three non-overlapping regions: high light, well-lit, and darkness. However, within well-lit regions, colored illumination still results in complex patterns with similar intensity levels, which require further subdivision. Therefore, we derive an invariant to extract features related to colored illumination and propose categorizing and disentangling patterns into darkness, well-lit, light effects, and high-light.
>
> We hope this explanation address your concerns.
>
> ***Q3: There is no baseline network, making it difficult to determine the performance gain for the specific module.***
>
> **R3:** The baseline network is the CUT [2], which operates without any physical information guidance in contrastive image translation. This baseline is included in the performance comparison in the Table 1, showing an improvement of 13.8 in FID scores and 9.84 SIFT scores on the Alderley dataset. Additionally, it shows an improvement of 24 in FID scores and 12.25 in mIoU on the BDD100k dataset compared to the baseline. Our ablation study in Table 2, Section 4.4, identifies the effectiveness of each module.
> We promise to highlight these improvements in the main paper for better understanding.
>
> [2] Contrastive learning for unpaired image-to-image translation. T. Park. A. A. Efros, R. Zhang, J. Y. Zhu. ECCV 2020.
>
> ***Q4: For instance, there are no ablation experiments to verify the impact of introducing four regions for disentanglement versus three regions (e.g., excluding the light effects region).***
>
> **R4:** We have provided this analysis in Section 4.4, as shown in the second table of Table 2. When only $L$ is activated, it indicates that we are only incorporating darkness, well-lit, and high light for disentangling.  When only $L$ and $N$ both are activated, it represents the full method. The full method demonstrates over a 10-point improvement in FID performance compared to using only three types of disentanglement across both datasets. We promise to clarify this expression for better understanding.
>
> ***Q5: There is a need to compare with more recent methods for unpaired image-to-image translation, such as COCO-FUNIT, StegoGAN, GP-UNIT, etc.***
>
> **R5:** We provide a comparison on the two datasets with the most advanced method, StegoGAN (CVPR 2024). It is clear that our method consistently outperforms StegoGAN, demonstrating superior performance.  We will include this in the camera-ready version.
> |                    |BDD100k|          | Alderley|           |
> | :-----------     | :------:  | :------:  | :--------:    | :------:  |
> |                    |FID         |LPIPS      |  FID       |LPIPS     |
> |StegoGAN   |   89.9     | 0.687      | 82.8       |    0.718  |
> |Ours            | **31.5**  |**0.466** |**50.9** |**0.650**|
>
> ***Q6: Please check the reference [5], as it does not seem to be published in CVPR.***
>
> **R6:** This paper was published in the CVPR Workshop 2023. We will revise this in the main paper. Thank you for your reminder.
>
> ***Q7: How is the reweighting matrix obtained through optimal transport? Are there any references that have implemented this approach?***
>
> **R7:** The reweighting matrix is obtained by solving the optimization problem in Eq (14). A similar approach is employed in MoNCE (CVPR 2022), which is also compared in our experiments. Unlike MoNCE, which computes the reweighting matrix across entire image patches, our method designs the reweighting matrix specifically for each degradation regions. This tailored approach results in over 20 FID score improvements on the Alderley dataset and nearly 10 FID score improvements on the BDD100K dataset compaired to MoNCE, as shown in Table 1.
>
> ***Q8: Contrastive learning has been widely used in image translation. The paper lacks ablation experiments to support the performance gains from the reweighting operation.***
>
> **R8:** We discuss the performance gains from the reweighting operation in the first subtable of Table 2 in Section 4.4 of the main paper. The notation (b) represents the degradation-aware reweighting operation. This table shows nearly a 5 FID score improvement from the reweighting operation on both two datasets.

---

> ### Author Response · Authors · 2024-08-13
>
> Dear Reviewer **rJuz**,
>
> Thank you for taking the time to review our submission and for your constructive comments and favorable recommendation. We would like to confirm whether our responses have adequately addressed your earlier concerns. If you have any additional questions or suggestions, we would be happy to address them to further enhance the quality of our paper.
>
> Best regards,
>
> Authors

---

### Official Review · Reviewer_JHwd · 2024-07-16

**Soundness:** 3
**Presentation:** 2
**Contribution:** 2
**Rating:** 5
**Confidence:** 4

**Summary:**

This paper proposes to address the night-to-day translation problem in which its learning basically can be briefly described by two steps: 1) illumination distribution as well as the physic priors built upon the Kubelka-Munk photometric model are firstly adopted to separate/disentangle the image regions into four degradation categories, i.e. darkness, well-lit, light effects, and high-light, in which such illumination degradation disentanglement is the main contribution of the proposed method; 2) the degradation-aware contrastive learning module is applied to maximize the mutual information between patches in the same spatial location from the generated image and the source image, where the anchor and its corresponding negative patches should be from the same degradation category (i.e. degradation-aware sampling) and the weights for each negative patch are determined by similar matrix obtained from the optimal transport computation (i.e. degradation-aware reweighting). Moreover, the GAN-based objective function is employed to bridge the domain gap between (generated) daytime and nighttime images. The translated images (from nighttime to daytime) are shown to have better quantitative and qualitative performance (in terms of FID) for aligning with the real nighttime image distribution.

**Strengths:**

+ In addition to provide better translation performance (both quantitative and qualitative), the translated images produced by the proposed method are shown to have better structural similarity with respect to the corresponding daytime images (evaluated in Alderley dataset)  in comparison to the baselines. Moreover, the typical semantic segmentation model (pretrained on typical daytime dataset, i.e. Cityscapes) applied on the translated images (produced by the proposed method) leads to better segmentation results in comparison to being applied on the images generated by the baselines (i.e. indirect evidence showing that the images generated by the proposed better follows the daytime image distribution which the semantic segmentation model is trained on).
+ The ablation study does demonstrate the contribution of illumination degradation disentanglement for separating the image patches into four different degradation categories.

**Weaknesses:**

- Although experimentally shown to be effective, the mechanism and the basic ideas behind leveraging the illuminance distribution as well as the physic priors for realizing disentanglement of four degradation categories (i.e. darkness, well-lit, light effects, and high-light) are not well explained, in which the physical meanings for Eq.1 to Eq.11 are hard to understand and follow. Basically, as such illumination degradation disentanglement is the main contribution of the proposed method, the description should be more self-contained and explanatory.
- As the illumination degradation disentanglement plays a key role in the proposed method, it would be great to have the robustness analysis on such disentanglement if possible (i.e. how accurate is the disentanglement, is there any related dataset we could apply such analysis?) and how would the translation model learning be affected once there are erroneous disentanglement?

**Questions:**

Though the contribution and the novelty of the proposed illumination degradation disentanglement is clearly recognizable from the quantitative and qualitative results, better description upon its mechanism and basic ideas would be much appreciated. Moreover, the further analysis upon its accuracy/robustness would be also better to have.

**Limitations:**

no potential negative societal impact is found.

---

> ### Author Rebuttal · Authors · 2024-08-06
>
> ***Q1:  Although experimentally shown to be effective, the mechanism and the basic ideas behind leveraging the illuminance distribution as well as the physic priors for realizing disentanglement of four degradation categories (i.e. darkness, well-lit, light effects, and high-light) are not well explained, in which the physical meanings for Eq.1 to Eq.11 are hard to understand and follow.***
>
> **R1:** Thanks for your valuable feedback.
>
> First, we will explain our basic ideas behind leveraging the illuminance distribution as well as the physic priors for realizing disentanglement of four degradation categories.  At nighttime, the intensity of illumination is the most important criterion for determining whether patterns are far from each other, which categorizes nighttime images into three non-overlapping regions: high light, well-lit, and darkness. However, within well-lit regions, colored illumination still results in complex patterns with similar intensity levels, which require further subdivision. Therefore, we derive an invariant to extract features related to colored illumination and propose categorizing and disentangling patterns into darkness, well-lit , light effects, and high-light.
>
>
>
> Second, we apologize for the omission of a key citation, which made it challenging to understand the derivations from Eq (1) to Eq (11). Specifically:
> + Eq (1) is derived from [1].
> + Eqs (2)–(4) present a simplified photometric model tailored for nighttime environments, proposed by us.
> + Eq (5) introduces the invariant from [1], and we derive new characteristics based on our simplified model in Eq (6).
> + Eqs (7)–(10) follow [1] for computing the basic invariant and refining it for disentangling.
> + Eq (11) employs the estimated light effect to refine the mask and extract light effect regions from the original well-lit regions.
>
> We will include these clarifications and the key citation in the revised version to improve understanding.
>
> [1] Color Invariance. J. M. Geusebroek, R. van den Boomgaard, A. W. M. Smeulders, H. Geerts. IEEE TPAMI, 2001.
>
> ***Q2:  As the illumination degradation disentanglement plays a key role in the proposed method, it would be great to have the robustness analysis on such disentanglement if possible and how would the translation model learning be affected once there are erroneous disentanglement?***
>
> **R2:** We agree that a robustness analysis of disentanglement is important. Unfortunately, conducting such research is nearly impossible due to the lack of illumination-related annotations. Creating such a dataset is also challenging due to the absence of unified measurement criteria in this field. Despite this, we provide a task-oriented evaluation of different disentanglement strategies by comparing final performance in the second subtable of Table 2 in the main paper. This table demonstrates that the 4 types of degradation disentanglement leads to significant performance improvements compared to the initial 3 types, indicating that the proposed 4 types disentanglement is empirically more reasonable.
>
> We are committed to advancing this area and will continue to work on robustness analysis, including proposing measurement standards to make the field more complete.

---

> ### Author Response · Authors · 2024-08-13
>
> Dear Reviewer **JHwd**,
>
> Thank you for taking the time to review our submission and for your constructive comments and favorable recommendation. We would like to confirm whether our responses have adequately addressed your earlier concerns. If you have any additional questions or suggestions, we would be happy to address them to further enhance the quality of our paper.
>
> Best regards,
>
> Authors

---

### Author Rebuttal · Authors · 2024-08-06

Thank you to all the reviewers for meticulously evaluating our paper.

First of all, we promise to add the missing citation in the revised version and apologize for our oversight. While we employ the color invariant from [1], we are the first to discuss its characteristics in nighttime scenes and identify its potential as a light effect detector to disentangle illumination degradations, both theoretically and empirically.

Next, we will provide a detailed explanation of the motivation behind the four degradation types to address common concerns.

A key observation is that, when perform Night2Day translation,  treating all regions equally leads to significant artifacts by mixing different patterns from various regions. For example, mixing light effects patterns with high-light patterns, as shown in the Figure 1 of the main paper. Intuitively, separating these patterns and regularizing the structure will help mitigate these artifacts.


Based on this intuition, we sought a disentanglement strategy to separate these patterns. At nighttime, the intensity of illumination is the most important criterion for determining whether patterns are far from each other, which categorizes nighttime images into three non-overlapping regions: high light, well-lit, and darkness. However, within well-lit regions, colored illumination still results in complex patterns with similar intensity levels, which require further subdivision. Therefore, we derive an invariant to extract features related to colored illumination and propose categorizing and disentangling patterns into darkness, well-lit, light effect, and high light to address these challenges.

We agree that this disentanglement could be more precise. However, achieving this precision solely through physical priors is challenging and may require annotations for these degradation types and well-trained segmentation models.

Additionally, we provide a one-page PDF file showing additional experimental results including :
+ Ablation visualizations with the initial 3-cluster method and the full disentanglement method.
+ Segmentation visualizations of the proposed methods.

Thanks again to all the reviewers.


[1] Color Invariance. J. M. Geusebroek, R. van den Boomgaard, A. W. M. Smeulders, H. Geerts. IEEE TPAMI, 2001.

---

### Decision · Program_Chairs · 2024-09-25

**Decision:**

Reject

**Comment:**

This paper presents a new approach to night-to-day image translation by introducing a degradation disentanglement module and degradation-aware contrastive learning. The core contribution of this paper lies in categorizing nighttime images into four degradation categories—darkness, well-lit, light effects, and high light—guided by physical priors from the Kubelka-Munk photometric model. The method demonstrates superior performance on the BDD100K and Alderley datasets, particularly in terms of FID scores and downstream task performance.

The paper received 4 reviews with diverging paper ratings: 2 Borderline Rejects and 2 Borderline Accepts. The reviewers' opinions are divided. Some acknowledge the novelty and potential impact of the work, while others point out the need for significant revisions. The paper is technically solid but has several issues related to clarity, presentation, and depth of technical explanations. Given the need for substantial revisions to improve the paper's clarity, presentation, and technical soundness, the AC agrees with reviewers rJuz and dsJ3 and suggest a reject for this paper.